# Effect of environmental variables on the abundance of *Amblyomma* ticks, potential vectors of *Rickettsia parkeri* in central Brazil

Isadora R. C. Gomes[1,2,☯], Rodrigo Gurgel-Gonçalves[2,☯], Gilberto S. Gazeta[3]\*, Ana B. P. Borsoi[3], Karla Bitencourth[3], Letícia F. Leite[1], Nathália G. S. S. Coelho[1], Ricardo Dislich[4], Helga C. Wiederhecker[5], Eduardo G. Santos[6], Melina Guimarães[1]

**1** Biological Sciences Course, Catholic University of Brasília, Federal District, Brasília, Brazil, **2** Faculty of Medicine, Laboratory of Medical Parasitology and Vector Biology, University of Brasília, Federal District, Brasília, Brazil, **3** National Reference Laboratory for Rikettsiosis Vectors, Oswaldo Cruz Foundation, Rio de Janeiro, Brazil, **4** Ministry of Planning and Budget, Esplanade of Ministries, Federal District, Brasília, Brazil, **5** Independent Researcher, Federal District, Rio de Janeiro, Brazil, **6** Post Graduation Program in Ecology, Institute of Biological Sciences, University of Brasília, Federal District, Brasília, Brazil

☯ These authors contributed equally to this work.
\* gsgazeta@ioc.fiocruz.br

**Data Availability Statement:** All relevant data are within the manuscript and its Supporting Information files.

## Abstract

*Amblyomma* ticks are vectors of both *Rickettsia rickettsii* and *R. parkeri* in the Americas, where capybaras (*Hydrochoerus hydrochaeris*) are the main hosts in urban areas, thus contributing to the transmission of spotted fever. Herein, we studied: (i) the seasonal dynamics and abundance of ticks in areas where capybaras live, (ii) the effect of environmental variables on tick abundance, and (iii) the presence of *Rickettsia*-infected ticks. Between September 2021 and September 2022, we sampled ticks using cloth-dragging at 194 sites on the shore of Lake Paranoá in Brasília, Brazil. We measured environmental data (season, vegetation type, canopy density, temperature, humidity, and presence or vestige of capybara) at each site. Nymphs and adults were morphologically identified to the species level, and a selected tick sample including larvae was subjected to genotypic identification. We investigated *Rickettsia*-infected ticks by PCR (*gltA*, *htrA*, *ompB*, and *ompA* genes) and associations between tick abundance and environmental variables using Generalized Linear Models. A total of 30,334 ticks (96% larvae) were captured. Ticks were identified as *Amblyomma*, with *A. sculptum* comprising 97% of the adult/nymphs. Genotype identification of a larval sample confirmed that 95% belonged to *A. dubitatum*. Seasonal variables showed significant effects on tick abundance. Most larvae and nymphs were captured during the early dry season, while the adults were more abundant during the wet season. Vegetation variables and the presence of capybaras showed no association with tick abundance. *Rickettsia parkeri* group and *R. bellii* were identified in *A. dubitatum*, while *A. sculptum* presented *R. bellii*. We conclude that: (i) *Amblyomma* ticks are widely distributed in Lake Paranoá throughout the year, especially larvae at the dry season, (ii) the abundance of *Amblyomma* ticks is explained more by climatic factors than by vegetation or presence of capybaras, and (iii) *A. dubitatum* ticks are potential vectors of *R. parkeri* in Brasília.

**Funding:** IRCG and EGS received specific funding of Coordenação de Aperfeiçoamento de Pessoal de Nível Superior https://www.capes.gov.br/ Award Number: finance code 001. RGG received funding from the National Council for Scientific and Technological Development (CNPq, Brazil, award number 314892/2021-4). The funding sources of this study had no role in the study design, data collection, data analysis, data interpretation, writing of the report, or in the decision to submit the paper for publication.

**Competing interests:** The authors have declared that no competing interests exist.

**Abbreviations:** GLM, generalized linear models; PCR, Polymerase Chain Reaction.

# Introduction

*Amblyomma* ticks (Acari: Ixodidae) are vectors of both *Rickettsia rickettsii* and *R. parkeri* in the Americas, where capybaras (*Hydrochoerus hydrochaeris*) are the main hosts in urban areas, thus contributing to the transmission of spotted fever rickettsiosis [1–4]. *Amblyomma* ticks frequently infest capybaras [5,6], and *Amblyomma sculptum* Berlese 1888 is the most important vector of Brazilian Spotted Fever (BSF), a severe disease caused by *R. rickettsii* [7–10]. A milder form of human rickettsiosis caused by *R. parkeri* has also been reported in Brazil [1].

Between 2007 and 2021, Brazil has reported around 36,500 suspected cases of BSF, with 7% being confirmed, resulting in approximately 170 cases annually and a total of 837 deaths [11]. Reports have shown most cases in the southern and southeastern regions [12,13]. However, several studies have found ticks, capybaras, and other animals infected with *Rickettsia* in the central-western region [14,15]. In Brasília, one confirmed case of BSF was reported [11], and a study of ticks associated with capybaras on the shore of the Lake Paranoá found *A. dubitatum* Neumann 1899 infected with *Rickettsia parkeri*-like agent [15]. Additionally, 53 out of 55 capybara serum samples tested by indirect immunofluorescence reaction were positive for *Rickettsia*, of which 21 have antigen of *Rickettsia bellii*. Moreover, PCR revealed *R. bellii* in 25 out of 108 (23.1%) tick samples collected from capybaras [15].

Although previous studies support that *Amblyomma* ticks are infected with *Rickettsia* in our study area, the factors that influence tick abundance and infection in areas occupied by capybaras in Brasília are unknown. Several factors may influence tick abundance and infection in urban areas, such as vegetation density [16], climate [17], host availability [4,18–21], human behavior [22,23], and control measures [24,25]. Then, it is necessary to understand ecological factors influencing tick abundance and infection in urban areas where capybaras occur to estimate the potential risk of *Rickettsia* transmission to humans and to guide control measures, contributing to the surveillance of BSF. Herein, we studied: (i) seasonal dynamics and abundance of ticks in areas where capybaras live in Brasília, (ii) the effect of environmental variables on tick abundance, and (iii) the presence of *Rickettsia*-infected ticks.

# Materials and methods

## Study area

Lake Paranoá covers 37.5 km$^2$ in an environmental protection area in Brasília [26] (Fig 1). The lake shore vegetation consists of a mixture of exotic vegetation, cerrado (Brazilian savanna) remnants, and gallery forests [27]. The region is classified as having a tropical savanna climate with a dry winter. The seasonality is marked, with a dry season from May to September and a wet season from October to April [28]. Brasília has an average annual rainfall of 1500 to 1800 mm [29]. Lake Paranoá is an artificial lake designed to increase the relative humidity of the air in Brasilia. It serves as an important leisure spot for the population of Brasília. The shores of the lake offer restaurants, entertainment, and sports facilities.

## Tick collection and identification

We used the dragging technique [30] to collect ticks. We sampled different sites around the Lake Paranoá during six periods (variable *season*): September 2021, November 2021, February/March 2022, April 2022, June 2022, and September 2022. We randomly chose sampling sites and did all collections in the morning. To access the sampling sites near the lake shore, we used a motorboat. Our sampling effort ranged from 5 to 6 days per period.

We used soft and terry cloth (50cm x 60cm) attached to cleaning rods to collect ticks. We dragged the rod in a delimited area of 6m x 10m. In each site, we utilized twelve pieces of cloth

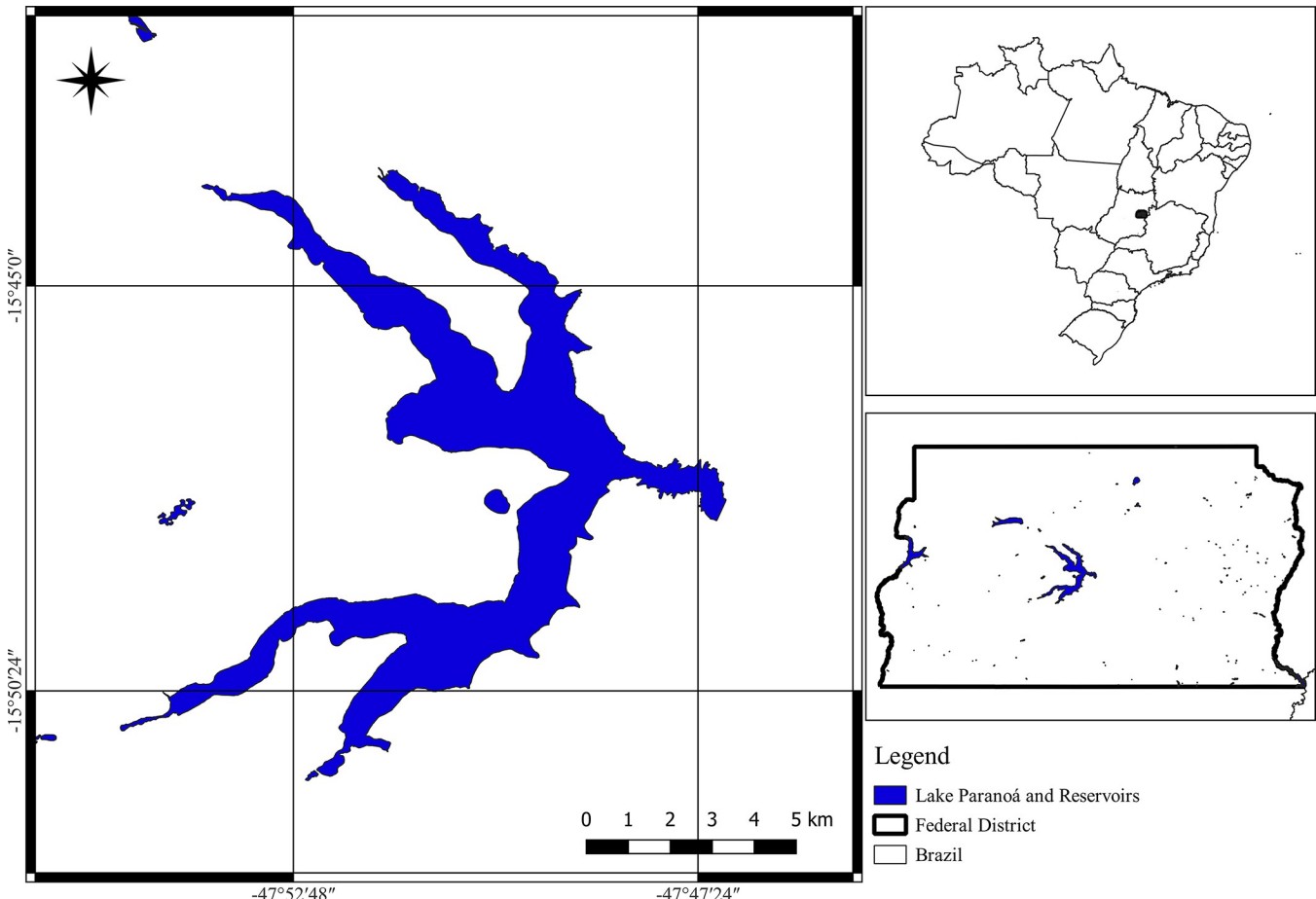

**Fig 1. Study area on Lake Paranoá in Brasília, Federal District of Brazil.** Shapefile: Instituto Brasileiro de Geografia e Estatística (IBGE–URL: https://www.ibge.gov.br/) and Instituto de Pesquisa e Estatística do Distrito Federal (IPEDF—URL: https://catalogo.ipe.df.gov.br/layers/geonode_data:geonode:Lagos_e_reservatorios).

(six made of soft material and six made of terry cloth) to prevent losing ticks during the process (Fig 2). The collected ticks were preserved on-site in 70% alcohol, stored in previously identified sealed plastic bags, and transported to the laboratory for identification under a stereomicroscope.

### Tick identification and *Rickettsia* research

Each tick was assigned to one of three life stages: larva, nymph, or adult [5]. Larvae were identified at the genus level using external morphology, and nymphs and adults were identified at the species level using printed keys [3,5,31–33]. For confirmation/specific identification of ticks and *Rickettsia* research, 1,756 individuals collected from September 2021 to April 2022 were analyzed either individually (adults) or in pools of 5 nymphs and 10 larvae, separately. To extract genomic DNA (gDNA) from tick samples, we used the saturated sodium chloride solution protocol [34]. We identified the genotypes of these tick samples by using PCR amplification of a fragment of the mitochondrial 16S rDNA gene [35].

*Rickettsia* was found in the ticks by amplifying fragments of *gltA*—citrate synthase [36] and *ompA*—outer membrane protein A [37] in a screening PCR. Afterwards, we amplified fragments of the *htrA* gene, which encodes the 17-kilodalton structural membrane protein (nested

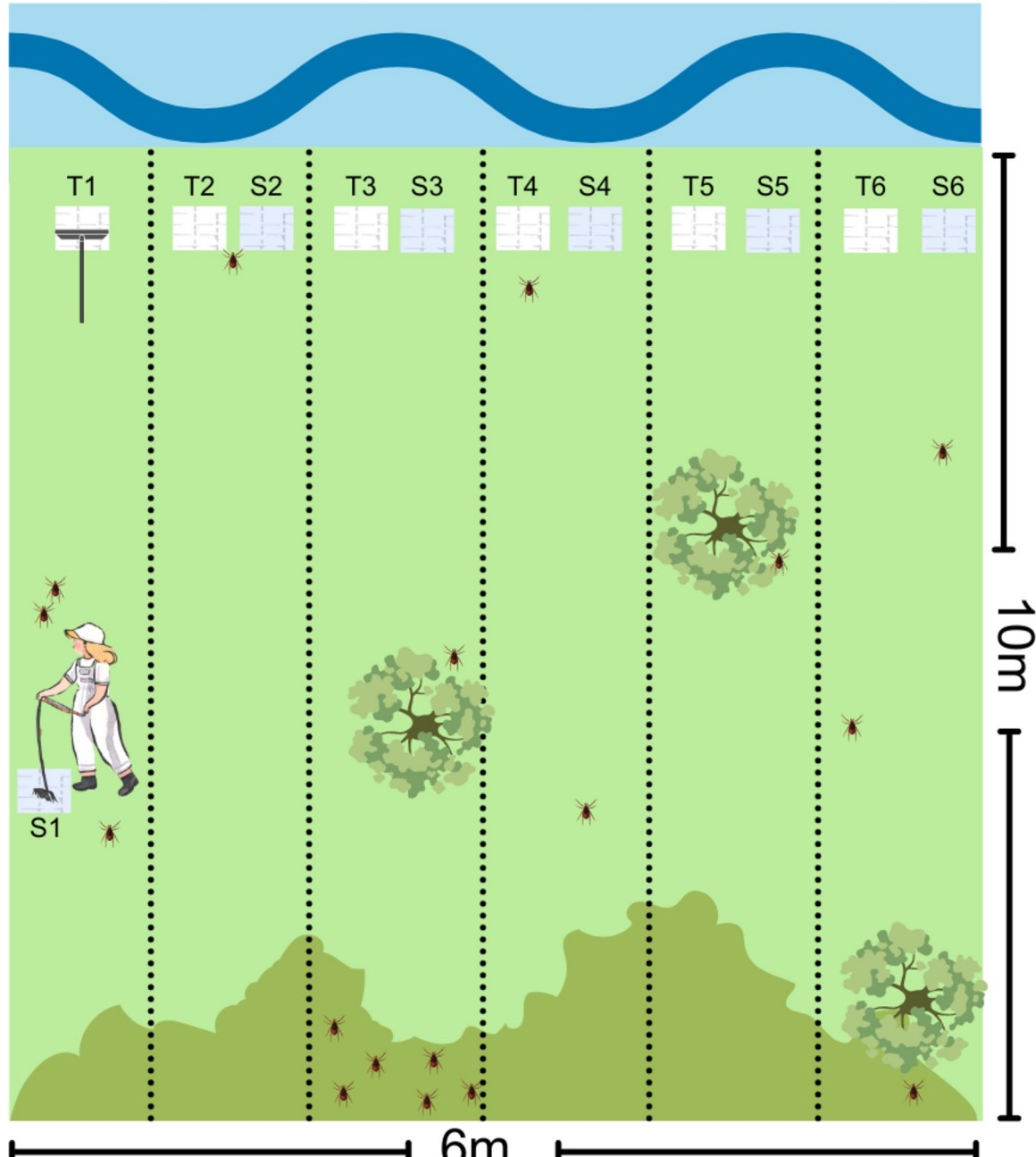

**Fig 2. Sampling of ticks on the shore of Lake Paranoá, Brasília, Federal District, Brazil, 2021/2022.** S: Soft cloth. T: terry cloth. The picture displays dark green bushes, trees, and ticks on different parts of the landscape.

PCR) [38,39], and the *ompB* gene, which encodes the outer membrane protein B (nested PCR) [40] on positive samples. Each test used 300 ng of *R. rickettsii* DNA as a positive control and ultrapure water without DNase and RNase as a negative control. We carried out all PCR reactions using a GeneAmp PCR System 9700 thermocycler (Applied Biosystems®, Carlsbad, USA).

PCR products were visualized by electrophoresis in 2% agarose gel stained with ethidium bromide and purified using the Wizard® SV Gel and PCR Clean-Up System Kit

(PromegaCorp., Madison,WI,USA) according to the manufacturer's protocols. DNA was sequenced in both directions using the same PCR primers and the BigDye Terminator™version3.1 Cycle Sequencing® Kit (Applied Biosystems, Foster City, CA, USA) on an automated ABI3730xl DNA analyzer (Applied Biosystems, Foster City, CA, USA). All obtained sequences are available on GenBank (accession numbers OR760224- OR760264; OR767220- OR767227).

The sequences were assembled and edited using ChromasPro, version 1.5 (Technelysium Pty Ltd., Tewantin, QLD, Australia) and subjected to BLASTn analyses (https://blast.ncbi.nlm.nih.gov/Blast.cgi) to identify first their closest similarities to other organisms available in GenBank. For *Rickettsia* sequences, we performed multiple alignments with the ClustalW algorithm and manually verified them. Protein genes were turned into amino acids without any stop codons. A combined Maximum-Likelihood phylogeny (*gltA*+ *htrA*+ *ompB*) was determined using PhyML Software Version 3.0 [41]. The evolutionary model GTR+I indicated by MEGA 6.0 was used through BIC. Alignments were concatenated with Seaview Software [42].

## Environmental characterization

At each sampling site, the following environmental variables were recorded: vegetation type (*veg*), canopy density (*can*), temperature (*t*), relative air humidity (*rh*), and the presence or vestige of capybaras (*capves*), identified through their scat, beds, and tracks. We identified three types of vegetation: (1) grass; (2) shrubs and trees; and (3) exposed soil, grass, and shrubs. The measurement of *t* and *rh* was carried out with a multimeter (Instrutherm, Thal-300) and *can* was quantified using a convex spherical forest densitometer (Forestry Suppliers, model C).

## Capybara count

Due to the presence of capybaras in our region, a species usually associated with the presence of ticks [15,43], we mapped the presence of capybaras along the shore of Lake Paranoá during the study period. The capybaras were counted using a boat (an aluminum boat with an outboard motor) at about 20 km/h at approximately 30 m from the shore. These surveys were carried out monthly during the late afternoon (16:00 to 18:00). During the surveys, the presence of the capybaras and their geographical location were recorded.

## Statistical analysis

Statistical analyses were performed in the R statistical environment, version 4.2.3 [44], using the *Performance Analytics*, *pscl*, *car*, *lmtest* and *KernSmooth* packages. We ran exploratory analyses to test for variation in the number of ticks across life stages (larvae, nymphs, or adults), species, seasons, drag type, vegetation type, canopy density, temperature, relative air humidity, and presence or vestige of capybaras. To understand the relationship between the presence of capybaras and the abundance of ticks recorded, we created a kernel density map [45], using the KernSmooth package [46], with the records of capybaras for each month during the study. We then extracted the kernel intensity values (*capker*) for each tick collection site, according to the month of sampling. The map that describes the study area was generated using QGIS software version 3.28.13.

We analyzed the effect of environmental variables on tick abundances for each life stage by fitting Generalized Linear Models (GLMs), following the recommendations of Zuur et al. [47]. We ran an initial GLM including cloth type (soft or terry) as an explanatory variable and found no effect of this variable, so we used the total number of ticks (soft + terry) as the dependent variable in further analyses. The number of ticks is a count variable, indicating the use of a GLM with a Poisson error distribution and a log link function (Poisson GLM). In the case of moderate overdispersion ($1.5 < \varphi < 15$), we corrected the standard errors using a quasi-GLM

model in which the variance is given by the mean multiplied by the dispersion parameter $\varphi$ (quasi-Poisson GLM). In cases in which overdispersion in the data was high ($\varphi > 15$) and the frequency of zeros in the data was much higher than expected from a Poisson distribution, we used zero-inflated GLMs, "mixture" models in which the zeros are modeled as coming from two different processes: the binomial process (to model the probability of measuring a zero) and the count process, modeled by a Poisson (ZIP) or a negative binomial (ZINB) GLM [47]. In these cases, we used simple inflation models, where all zero counts have the same probability of belonging to the zero component. To assess if overdispersion was adequately considered, we compared corresponding ZIP and ZINB models by applying a likelihood ratio test.

The environmental explanatory variables considered were *veg*, *season*, *can*, *t*, *rh*, *capves* and *capker*. Data used are available in S1 Data. The collinearity among explanatory variables was analyzed using variance inflation factors (VIF) as implemented in the car package, considering 5 as the cutoff value (variables with VIF > 5 were excluded from the analysis). For each life stage, variable selection to find the optimal model was performed by applying backward procedures, systematically removing variables that did not contribute significantly (p > 0.05) to the model. For quasi-Poisson GLM, we used an appropriate analysis of deviance to compare nested models with the full model, and for ZINB, likelihood ratio tests were employed (S1 File).

### Ethics approval and consent to participate

The adopted procedures were in accordance with the ethical standards of the Research Ethics Committee of the Fiocruz (Rio de Janeiro, Brazil) and with the Helsinki Declaration of 1964, revised in 1975, 1983, 1989, 1996, and 2000. Access to the research site was authorized by the Department of Environment of the Federal District. Permission to collect ticks was granted by the Chico Mendes Institute for Biodiversity Conservation (ICMBio) through the Biodiversity Authorization and Information System (SISBIO), request number 77851, authentication code 0778510320220711.

### Results

We collected 30,334 ticks from 115 of 194 sampled sites. Most ticks were captured during the larval stage (Table 1). Morphological identification confirmed that adult/nymphs were *Amblyomma*, mainly *A. sculptum*; genotypic identification of larvae showed that 95% were *A. dubitatum* (Table 2).

Tick life stages were detected in all sampling periods. The larvae were the most abundant life stage; most of them were found in April and June 2022, during the early dry season. Nymphs were more abundant in June of 2022 during the dry season, while adults were more common in November of 2021 during the wet season (Fig 3).

During the initial analysis of larval abundance, a Poisson GLM revealed substantial overdispersion ($\varphi = 227.3$), leading to the use of zero-inflated models. A ZIP GLM including all the explanatory variables exhibited high VIF values ($> 14$) for *t* and *rh*. After excluding *rh*, the VIF for *t* was reduced to 6.72. However, due to persisting collinearity concerns, *t* was also excluded from further analyses. The optimal model was a ZINB GLM, which retained only the variable *season*. Jun2022 had a significantly higher larval count compared to sep2021 (p < 0.001), nov2021 (p < 0.001), feb2022 (p < 0.001), and sep2022 (p < 0.001). Variables *veg*, *can*, *capves* and *capker* showed no significant effect and were excluded from the optimal model during the selection procedure (S1 File).

For the analysis of nymphs, we identified moderate overdispersion ($\varphi = 9.92$), which required the use of a quasi-Poisson GLM. The optimal model retained the variables *season*, *t*,

**Table 1. Number of ticks captured on the Lake Paranoá shore in Brasília, Federal District of Brazil, between 2021 and 2022.**

| Month | Year | Cloth | Life stage | | | Total |
|---|---|---|---|---|---|---|
| | | | Larvae | Nymph | Adult | |
| September | 2021 | Terry | 556 | 117 | 4 | 677 |
| | | Soft | 121 | 62 | 4 | 187 |
| November | 2021 | Terry | 339 | 38 | 27 | 404 |
| | | Soft | 608 | 17 | 26 | 651 |
| February/March | 2022 | Terry | 207 | 0 | 10 | 217 |
| | | Soft | 332 | 1 | 14 | 347 |
| April | 2022 | Terry | 4124 | 52 | 15 | 4191 |
| | | Soft | 4883 | 39 | 8 | 4930 |
| June | 2022 | Terry | 11872 | 268 | 11 | 12151 |
| | | Soft | 5904 | 325 | 2 | 6231 |
| September | 2022 | Terry | 101 | 46 | 3 | 150 |
| | | Soft | 154 | 42 | 2 | 198 |
| | | Total | 29201 | 1007 | 126 | 30334 |

and *capves*; however, only the season variable showed a significant effect. Specifically, the data from jun2022 demonstrated a significantly higher number of individuals compared to sep2021 (p = 0.003), nov2021 (p < 0.001), apr2022 (p < 0.001), and sep2022. Variables *veg*, *can*, *rh* and *capker* showed no significant effect and were excluded from the optimal model during the selection procedure (p < 0.001) (S1 File).

For adult specimens, we observed moderate overdispersion ($\varphi$ = 1.84), which led to the adoption of a quasi-Poisson GLM. The optimal model indicated a negative effect of *rh* (p = 0.015) and an effect of *season*, with nov2021 having a significantly higher number of individuals compared to sep2021 (p = 0.002), apr2022 (p = 0.048), jun2022 (p = 0.006), and sep2022 (p = 0.002). Variables *veg*, *can*, *t* and *capves* showed no significant effect and were excluded from the optimal model during the selection procedure (S1 File). The optimal model included the variable *capker*, but it did not show a significant effect (p = 0.092). Moreover, no clear agreement was observed when comparing the heat map of capybara distribution in Lake Paranoá throughout the year with the distribution of ticks captured in the same areas; low capybara densities were observed in some areas with high tick abundance (Fig 4).

We tested 254 pool tick samples for *Rickettsia*, of which 37 (14.5%) were positive (in all months, mainly in November 2023, S1 Table) for the *gltA* gene, indicating the presence of

**Table 2. Tick species captured on the shore of Lake Paranoá, Brasília, Federal District of Brazil, between 2021 and 2022, by species and life stage.**

| Month | Year | *Amblyomma spp.* | *Amblyomma sculptum* | | *Amblyomma dubitatum* | | Total |
|---|---|---|---|---|---|---|---|
| | | Larvae | Nymph | Adult | Nymph | Adult | |
| September | 2021 | 592 | 166 | 9 | 13 | 0 | 780 |
| November | 2021 | 947 | 45 | 56 | 6 | 0 | 1054 |
| February/March | 2022 | 550 | 0 | 26 | 1 | 1 | 578 |
| April | 2022 | 5897 | 60 | 22 | 5 | 1 | 5985 |
| June | 2022 | 17776 | 582 | 3 | 11 | 0 | 18372 |
| September | 2022 | 255 | 101 | 5 | 0 | 0 | 361 |
| | Total | 26017[a] | 954 | 121 | 36 | 2 | 27130[b] |

[a] 95% of the larvae sample was genotyped as *A. dubitatum*

[b] Values in Tables 1 and 2 differ due to loss of samples during transit to the National Reference Laboratory for Rickettsiosis Vectors in Rio de Janeiro.

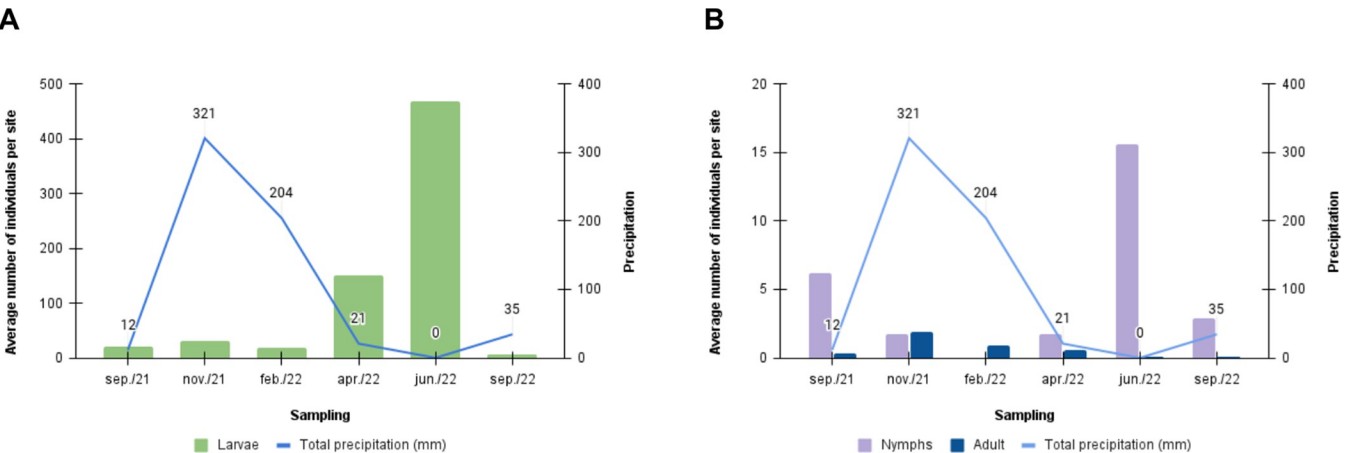

**Fig 3. Average tick abundance. A**, larvae; **B**, nymphs and adults. Number of individuals per site found on the shores of Lake Paranoá in the months of September and November 2021, and February, April, June and September 2022, and total rainfall for the collection periods.

bacteria of the genus *Rickettsia*. The positive samples were from larvae (mainly *A. dubitatum*), nymphs (4 *A. dubitatum* and 1 *A. sculptum*), and adults (1 *A. sculptum*). Three of these pools were also amplified for the *htrA* and *ompB* genes (September), indicating the presence of bacteria of the spotted fever group (SFG) (S1 Table). The concatenated phylogeny of the *gltA*, *htrA*, and *ompB* rickettsial gene fragments showed detection of *Rickettsia* infection in *Amblyomma* samples. The ompA gene fragment was not amplified. Three larvae samples of *A. dubitatum* (01G, 10D and 10E, S1 Table) were detected infected with the *R. parkeri* group. Additionally, 34 pools were identified with *R. bellii* (all the obtained sequences were 100% similar and classified as Haplotype I) (Fig 5).

## Discussion

Ticks collected from the shore of Lake Paranoá in Brasília all belonged to the *Amblyomma* genus, specifically *A. sculptum* and *A. dubitatum*. During the dry season, larvae and nymphs were more abundant, while adult ticks were primarily detected in the wet season. Vegetation variables available at the shore of Lake Paranoá showed no association with tick abundance. Surprisingly, the presence or vestige of capybaras showed no effect on the number of ticks of all life stages. *Rickettsia* sp. was detected in 14.5% of the samples analyzed. Of these, 92% were identified as *R. bellii* and 8% as *R. parkeri* group. The exclusive presence of *R. parkeri* in *A. dubitatum* ticks suggests that this tick species may be the potential vector of this spotted fever group bacteria in Brasília.

Cloth-dragging is a commonly used and cost-effective technique for sampling ticks [48]. In this study, we chose this method due to its convenience, considering the high number of sites visited during each sampling period. To maximize tick collection, we modified the standard technique by using clothes with different textures and drag multiple smaller cloth pieces on the vegetation. This method has proven effective for dragging over the dense vegetation of Cerrado, where a large cloth would tear easily. Our findings showed that both soft and terry cloth are suitable for collecting ticks of different life stages without distinction. Nonetheless, we noted a considerable number of absences throughout all sampling periods. Combining dragging with other collection methods, such as visual inspection and $CO_2$ traps, has been noted as a viable way to maximize tick collection [30,49]. However, Ramos *et al.* [50] found that dragging with soft cotton cloth was more efficient than visual inspection and $CO_2$ traps. Moreover,

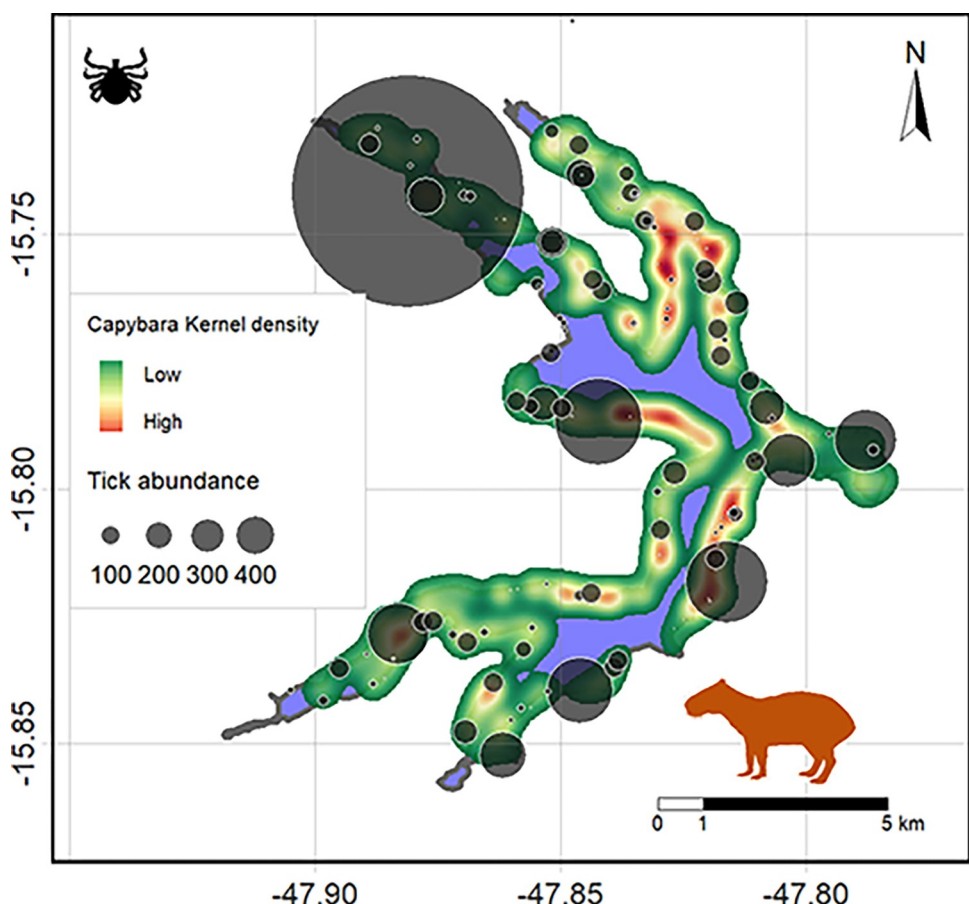

**Fig 4. Distribution of capybara and *Amblyomma* ticks at Lake Paranoá, Brasilia, Brazil.** The heat map represents the kernel density based on the groupings of capybaras observed during the 12 months of sampling. Warm colors represent places where capybaras are more abundant. The size of the circles represents the abundance of ticks counted during the study. Shapefile: Instituto Brasileiro de Geografia e Estatística (IBGE–URL: https://www.ibge.gov.br/).

Queiroz [51] observed that dragging is more efficient for sampling *A. sculptum* due to its stalking behavior on vegetation, while the use of $CO_2$ traps would be more appropriate for sampling *A. dubitatum* due to its attacking behavior. We believe that the use of other sampling methods in addition to the drag technique would allow us to collect a greater richness and abundance of species. Additionally, some collection points produced clusters of larvae, which could have introduced bias into our study due to the likelihood that these larvae belong predominantly to one species.

We found a higher abundance of *A. sculptum* adults and nymphs compared to *A. dubitatum*, in contrast to the results of Quadros *et al.* [15] in the same location, who found a higher abundance of *A. dubitatum* compared to *A. sculptum* in capybaras. This disparity may be because Quadros *et al.* [15] collected ticks directly from capybaras rather than from the vegetation. The sampling-effect hypothesis is supported by Paula *et al.* [52], who sampled ticks from the vegetation using dragging, flagging, and visual search. They found a lower frequency of *A. dubitatum* compared to *A. sculptum* in the Federal University of Goiás campus, where capybaras, coatis, cattle, and dogs are present. Queirogas *et al.* [18] suggest that *A. dubitatum* is more common in areas near water, associated with the presence of capybaras, while *A. sculptum* prefers drier areas further from the shore. This provides new insight and nuance to

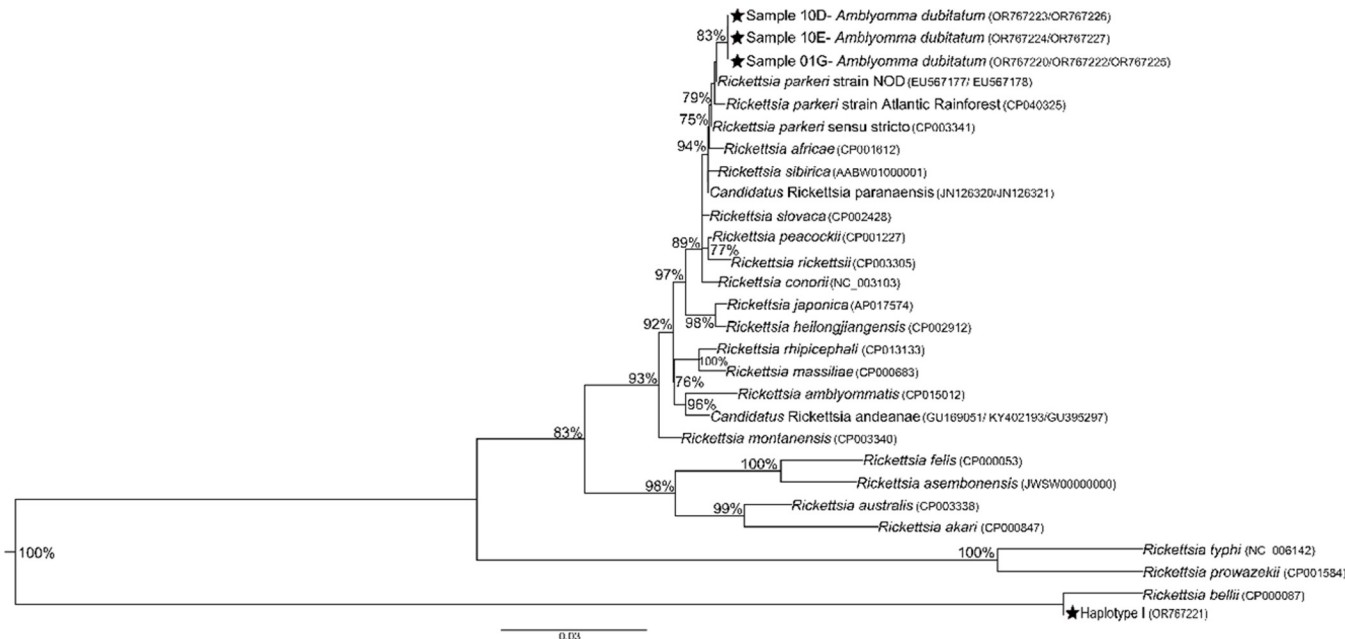

**Fig 5. Phylogenetic results of *Rickettsia* spp.** Concatenated phylogeny of the *gltA*, *htrA*, and *ompB* rickettsial gene fragments (380+409+403 bp) detected in *Amblyomma* samples in this study, inferred by maximum likelihood analysis with the GTR+I evolutionary model. The numbers on the branches represent the support values (70% cut-off). Stars (★) indicate sequences obtained in this study.

Queirogas *et al.* [18] proposal, suggesting that *A. dubitatum*, although associated with capybaras, is more restricted to less harsh conditions than *A. sculptum*. The effect of vegetation characteristics on tick abundance has been reported previously. In a study analyzing *Ixodes scapularis* using flag/drag techniques within a forested environment, Ginsberg *et al.* [53] discovered that canopy cover was a more reliable predictor of tick abundance than other vegetation characteristics. These results suggest that the effect of vegetation on tick abundance may depend on the species and the ecosystems in which they live; our results indicate that the vegetation variables we measured on the shores of Lake Paranoá do not significantly influence the abundance of *Amblyomma* ticks.

The seasonal influence on tick abundance observed in Lake Paranoá has also been reported in the literature. High abundance of larvae/nymphs of *Amblyomma* ticks has been described during the dry season in a riparian forest in São Paulo, an ecotone region between Atlantic Forest and Cerrado [54], as well as in other parts of São Paulo State [55] and in Cerrado areas of Goiás [50]. Furthermore, the high number of adults during the wet season is consistent with the seasonal dynamics described for *A. sculptum* in the southern, southeastern, and central-western regions of Brazil [4,52,56,57]. In these regions, larvae are abundant between April and July, nymphs between July and October, and adults between November and February. The rainfall pattern and climatic conditions that imply high temperature and humidity may be responsible for the rapid completion of the tick life stage and increase in adult population [54,58].

Our GLM results revealed that all life stages showed no significant effect of presence or vestiges of capybara. This was an unexpected result considering that *A. sculptum* and *A. dubitatum* are commonly associated with capybaras [18,59]. The presence of adult ticks (*A. dubitatum* and *A. sculptum*) on capybaras has been reported in our study area [15], suggesting that capybaras are hosts for the maintenance of the life cycle of *Amblyomma* on the shores of Lake Paranoá. The association between *Amblyomma* ticks and capybaras was also observed by

Nunes *et al.* [25] in São Paulo. After the culling of capybaras, the number of both *A. dubitatum* and *A. sculptum* dropped to almost zero, and the low abundance of adult *A. sculptum* ticks coincided with relatively low capybara numbers. Queirogas *et al.* [18] also found a positive correlation between capybara and tick abundance; however, the tick species had an uneven distribution and environmental factors rather than host availability should influence tick abundance. One hypothesis to explain the lack of a positive association between tick abundance and capybara found here could be related to the diversity of food sources. *Amblyomma* ticks on the shore of Lake Paranoá may rely on other mammalian and avian hosts besides capybaras to complete their life cycle, such as *Cavia aperea*, *Dasyprocta azarae*, *Vanellus chilensis*, *Callithrix penicillata*, *Didelphis albiventris*, *Gracilinanus agilis*, domestic dogs and cats, all animals observed in the shore during sampling. To investigate this issue in depth, molecular techniques would ideally be used to analyze the food sources of the collected ticks [60]. In this sense, it is reasonable to consider other animals besides capybaras as responsible for maintaining the circulation of *R. rickettsii* in BSF-endemic areas. The shores of Lake Paranoá are occupied by a variety of species, ranging from residential and protected areas to clubs and parks, which are visited daily by humans and domestic animals. These animals may have occasional contact with *A. sculptum* and carry ticks into residences, highlighting the role of pets as possible carriers of BSF vectors into human settlements [61]. We emphasized that the number of *A. sculptum* ticks may be influenced by the number of domestic animals, especially dogs and cats [62,63], and therefore we recommend that further studies in the Lake Paranoá consider the frequency of domestic animals in addition to capybaras presence.

The presence of *R. bellii* in ticks collected directly from capybaras near recreational sites was previously observed in Brasília [15]. *R. bellii* is commonly found in ticks and is not pathogenic to humans [64,65]. However, *R. parkeri*, a pathogenic bacterium of the spotted fever group, has been reported in capybaras in Brasília [15]. *R. parkeri* has also been reported in southern and southeastern Brazil, associated with *Amblyomma ovale* in Mato Grosso do Sul [66], *A. tigrinum* in the Pampas [67], and *A. triste* in Minas Gerais [68]. Recently, a new focus of spotted fever caused by *R. parkeri* has been described in Rio de Janeiro. Martiniano *et al.* [69] detected *R. parkeri* in human skin and in *A. ovale* from a dog, expanding the known range of this rickettsial disease in Brazil.

The BSF-endemic areas were characterized by much higher tick burdens on both capybaras and in the environment, when compared to the BSF-nonendemic areas [8]. Additionally, there is no serologic evidence of *R. rickettsii* infection in capybaras found in BSF-nonendemic areas, where capybaras have been found to be infected with other *Rickettsia* species, particularly *R. bellii* which may be related to the dominance of *A. dubitatum* ticks in [8]. Our results support the suggestion of Quadros *et al.* [15] that the Federal District of Brazil is not currently endemic for BSF, because none of the samples of *A. sculptum* or *A. dubitatum* tested positive for *R. rickettsii*. The non-detection of *R. rickettsia* in Brasília may be due to a low infection rate of capybaras and ticks, as observed in other regions of Brazil, where capybaras show 0.05%-1.28% infection [7]. At the same time, the uneven distribution of tick species might implicate an unequal risk of tick-borne diseases within the same urban area [18]. *A. sculptum* is the most frequent human-biting tick in southeastern Brazil and is the most important vector of BSF [7]. Furthermore, *A. dubitatum* has been commonly found to parasitize capybaras in southeastern Brazil, without any direct impact on BSF-epidemiology [70,71]. In São Paulo, *A. dubitatum* was found to be heavily infected with *R. bellii* [8], and experimental studies showed that *A. dubitatum* infected with *R. bellii* was only partially refractory to *R. rickettsii* and unable to transmit *R. rickettsii* transovarially [72]. These findings suggest that if *A. dubitatum* becomes prevalent in a particular area, *R. rickettsii* may not be able to establish an effective infection in either *A. dubitatum* or *A. sculptum*. In our study, it seems that most of the larvae captured in

the environment were *A. dubitatum*, which could also explain why *R. rickettsii* was not detected in examined ticks.

The endemism of BSF in an area can be attributed to the size of the capybara population. In 2006, a study was conducted in São Paulo before the endemism of BSF, where 78 capybaras and few *Amblyomma* ticks were sampled (0.7 *A. sculptum*/trap and 3.3 *A. dubitatum*/trap); after 6 years, this area had become endemic for BSF, and there was an increase in the number of capybaras (~3 times higher) and ticks (33 *A. sculptum*/trap and 2.1 *A. dubitatum*/trap) [57]. The study suggests that the emergence of BSF is linked to the growth of the capybara population, which offers a food source for *A. sculptum*. Consequently, capybara populations in urban areas are associated with high environmental tick infestation. This, in turn, increases the risk of tick bites and pathogen transmission to humans. Our results highlight the importance of monitoring capybaras and ticks (*A. dubitatum* and *A. sculptum*) in Lake Paranoá to study their population dynamics [73] and infection thus providing relevant data for the surveillance of *R. rickettsii* in Brasilia and preventing outbreaks of BSF transmission, which have been observed in the southeastern region of Brazil [1,7,8,10].

Our study indicates that ticks are more influenced by climatic factors than by the presence of capybara. Therefore, prevention measures should focus on avoiding exposure to ticks in the study areas, especially during the dry season. This can be achieved through personal protective measures, such as wearing appropriate clothing, as well as by displaying signs warning residents of the presence of ticks, which are potential vectors of BSF. Health education campaigns can also be conducted to inform people about what to do if they find a tick and when ticks are most abundant in the area [74–76].

## Conclusions

We conclude that: (i) *Amblyomma* ticks are widely distributed in Paranoá Lake throughout the year; (ii) the abundance of *Amblyomma* ticks is explained more by climatic factors than by vegetation or presence of capybaras; (iii) *A. dubitatum* ticks are potential vectors of *R. parkeri* in Brasília. Based on tick infection data, there is a potential risk of *Rickettsia* transmission to humans in this non-endemic but vulnerable area.

## Supporting information

**S1 Table. *Rickettsia*-infected ticks from Lake Paranoá shore, Brasília, Federal District of Brazil, between 2021 and 2022.**
(XLSX)

**S1 Data. Data used in the GLM analyses to estimate the effect of environmental variables on tick abundances for each life stage.**
(XLSX)

**S1 File. Rcode used to estimate the effect of environmental variables on tick abundances for each life stage.**
(TXT)

**S1 Graphical abstract.**
(PPTX)

## Acknowledgments

We are grateful to Morgana Bruno, coordinator of the project to which this work is linked; Mariana Velasquez, Filipe Pereira, Vitor Costa, Matheus Rego, Larissa Ferreira, Mariana Cruz,

Ingrid Machado, Filipe Vieira Ataides and Rodrigo Lima Martins de Oliveira for their important contribution to this study; the Genomic Platform DNA Sequencing (PDTIS, Oswaldo Cruz Foundation) for support with sample sequencing. We also thank José Venzal for insightful comments on the manuscript.

## Author Contributions

**Conceptualization:** Isadora R. C. Gomes, Rodrigo Gurgel-Gonçalves, Helga C. Wiederhecker, Eduardo G. Santos, Melina Guimarães.

**Data curation:** Isadora R. C. Gomes, Rodrigo Gurgel-Gonçalves, Karla Bitencourth, Ricardo Dislich, Eduardo G. Santos, Melina Guimarães.

**Formal analysis:** Karla Bitencourth, Ricardo Dislich, Eduardo G. Santos, Melina Guimarães.

**Funding acquisition:** Gilberto S. Gazeta, Melina Guimarães.

**Investigation:** Isadora R. C. Gomes, Rodrigo Gurgel-Gonçalves, Gilberto S. Gazeta, Ana B. P. Borsoi, Karla Bitencourth, Letícia F. Leite, Nathália G. S. S. Coelho, Ricardo Dislich, Melina Guimarães.

**Methodology:** Isadora R. C. Gomes, Rodrigo Gurgel-Gonçalves, Gilberto S. Gazeta, Ana B. P. Borsoi, Karla Bitencourth, Letícia F. Leite, Nathália G. S. S. Coelho, Melina Guimarães.

**Project administration:** Melina Guimarães.

**Resources:** Rodrigo Gurgel-Gonçalves.

**Supervision:** Rodrigo Gurgel-Gonçalves, Melina Guimarães.

**Writing – original draft:** Rodrigo Gurgel-Gonçalves.

**Writing – review & editing:** Isadora R. C. Gomes, Rodrigo Gurgel-Gonçalves, Gilberto S. Gazeta, Ana B. P. Borsoi, Karla Bitencourth, Letícia F. Leite, Nathália G. S. S. Coelho, Ricardo Dislich, Helga C. Wiederhecker, Eduardo G. Santos, Melina Guimarães.

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
