## [Decision Letter · Decision Letter 0]

14 Feb 2024

PONE-D-24-02067Effect of environmental variables on the abundance of Amblyomma ticks, potential vectors of Rickettsia parkeri in central BrazilPLOS ONE

Dear Dr. Rodrigo Gurgel-Goncalves,

Thank you for submitting your manuscript to PLOS ONE. After careful consideration, we feel that it has merit but does not fully meet PLOS ONE’s publication criteria as it currently stands. Therefore, we invite you to submit a revised version of the manuscript that addresses the points raised during the review process.

**ACADEMIC EDITOR: **The manuscript needs some revisions depending on the reviewers comments

We look forward to receiving your revised manuscript.

Kind regards,

Shawky M Aboelhadid, PhD

Academic Editor

PLOS ONE

Journal Requirements:

4. We note that Figures 1 & 4 in your submission contain [map/satellite] images which may be copyrighted. All PLOS content is published under the Creative Commons Attribution License (CC BY 4.0), which means that the manuscript, images, and Supporting Information files will be freely available online, and any third party is permitted to access, download, copy, distribute, and use these materials in any way, even commercially, with proper attribution. For these reasons, we cannot publish previously copyrighted maps or satellite images created using proprietary data, such as Google software (Google Maps, Street View, and Earth). For more information, see our copyright guidelines: http://journals.plos.org/plosone/s/licenses-and-copyright.

a. You may seek permission from the original copyright holder of Figures 1 & 4 to publish the content specifically under the CC BY 4.0 license. 

Reviewers' comments:

Reviewer's Responses to Questions

**Comments to the Author**

1. Is the manuscript technically sound, and do the data support the conclusions?

Reviewer #1: Yes

Reviewer #2: Yes

2. Has the statistical analysis been performed appropriately and rigorously? 

Reviewer #1: Yes

Reviewer #2: Yes

3. Have the authors made all data underlying the findings in their manuscript fully available?

Reviewer #1: Yes

Reviewer #2: Yes

4. Is the manuscript presented in an intelligible fashion and written in standard English?

Reviewer #1: Yes

Reviewer #2: Yes

5. Review Comments to the Author

Reviewer #1: The paper is well written. The methods were sufficient to answer the questions and the discussion is in accordance with the results obtained.

I have just one comment:

Lines. 283-284. “...Surprisingly, the presence or vestige of capybaras showed no effect on the number of ticks of all life stages...”

The authors completely ignored the fact that Lake Paranoá (in Brasília) is surrounded by a highly urbanized matrix. Anyone who reads the manuscript and doesn't know Brasília might even think that the study was carried out on the edge of a peri-urban lake, which is definitely not the case. The lack of relationship between the number of capybaras and the abundance of ticks in the sampled region is most likely related to urbanization. Most likely, the number of ticks may be related more to the number of domestic animals (especially dogs and cats) than specifically to capybaras. It is true that capybaras are the primary hosts of A. dubitatum, however, the same cannot be said for A. sculptum. It has been demonstrated that A. sculptum also parasitizes dogs and cats, including ticks infected with Rickettsia (see: Mendes et al. 2019 and Campos et al. 2020). Lake Paranoá, in addition to being surrounded by luxurious mansions and farms, whose owners raise many dogs and cats, residents from other regions of Brasília take their pets to the place for leisurely walks. The presence of these domestic animals in the research region, in my opinion, should not have been ignored in the experimental design. As it is not possible to return to the data collection phase, I suggest that the authors discuss whether urbanization and the presence of dogs and cats in the sampled area would have any impact on the results achieved.

It is also necessary to improve the quality of the figures.

References:

Campos SDE, Cunha NC, Machado CSC, Nadal NV, Seabra Junior ES, Telleria EL, et al. Spotted fever group rickettsial infection in dogs and their ticks from domestic–wildlife interface areas in southeastern Brazil. Braz J Vet Parasitol 2020; 29(1): e020219. http://doi.org/10.1590/S1984-29612020012.

Mendes JCR, Kmetiuk LB, Martins CM, Canavessi AMO, Jimenez T, Pellizzaro M, Martins TF, Morikawa VM, Santos APD, Labruna MB, Biondo AW. 2019. Serosurvey of Rickettsia spp. in cats from a Brazilian spotted fever-endemic area. Rev Bras Parasitol Vet 28:713–721.

Reviewer #2: - IT IS A DETAILED STUDY INCLUDING INFORMATION ABOUT ABIOTIC AND BIOTIC FACTORS.

BUT SOME ISSUES ARE NOT CLEAR, FOR EXAMPLE, HOW MANY DAYS A MONTH WAS THE SAMPLING EFFORT?

WAS IT A MONTHLY SAMPLING?

OTHER QUESTIONS ARE MARKED DIRECTLY IN THE TEXT.

6. PLOS authors have the option to publish the peer review history of their article (what does this mean?). If published, this will include your full peer review and any attached files.

Reviewer #1: No

Reviewer #2: **Yes: **DARCI MORAES BARROS BATTESTI

---

## [Author Response · Author response to Decision Letter 0]

12 Mar 2024

ACADEMIC EDITOR

Authors: Formatting changes in the manuscript file have been highlighted in yellow

Authors: Access to the research site was authorized by the Department of the Environment of the Federal District. Authorization to collect ticks was granted by the Chico Mendes Institute for Biodiversity Conservation (ICMBio) through the Biodiversity Authorization and Information System (SISBIO), request number 77851, authentication code 0778510320220711. (Lines 206-210)

3. Submissions in which author-generated code underpins the findings in the manuscript. In these cases, all author-generated code must be made available without restrictions upon publication of the work. Please review our guidelines at https://journals.plos.org/plosone/s/materials-and-software-sharing#loc-sharing-code and ensure that your code is shared in a way that follows best practice and facilitates reproducibility and reuse.

Authors: The script code used in R has been made available as supplementary data. (line 202)

4. We note that Figures 1 & 4 in your submission contain [map/satellite] images which may be copyrighted. All PLOS content is published under the Creative Commons Attribution License (CC BY 4.0), which means that the manuscript, images, and Supporting Information files will be freely available online, and any third party is permitted to access, download, copy, distribute, and use these materials in any way, even commercially, with proper attribution. For these reasons, we cannot publish previously copyrighted maps or satellite images created using proprietary data, such as Google software (Google Maps, Street View, and Earth). For more information, see our copyright guidelines: http://journals.plos.org/plosone/s/licenses-and-copyright.

Authors: Figure 4 was generated using our data, and the raster file was obtained from MapBiomas Brasil, which is available free of charge. We re-generated Figure 1 because it was taken from Google's My Maps and is protected by copyright. The figure was created using QGIS version 3.28.13 software. The shapefiles were obtained from the Brazilian Institute of Geography and Statistics (IBGE) and the Institute of Research and Statistics of the Federal District (IPEDF). The raster with land use categorization was obtained from MapBiomas. All files used are now in the public domain. (line 277)

5. Please review your reference list to ensure that it is complete and correct. If you have cited papers that have been retracted , please include the rationale for doing so in the manuscript text, or remove these references and replace them with relevant current references. Any changes to the reference list should be mentioned in the rebuttal letter that accompanies your revised manuscript. If you need to cite a retracted article, indicate the article’s retracted status in the References list and also include a citation and full reference for the retraction notice.

Authors: Done. 

Reviewer #1: The paper is well written. The methods were sufficient to answer the questions and the discussion is in accordance with the results obtained.

I have just one comment:

Lines. 283-284. “...Surprisingly, the presence or vestige of capybaras showed no effect on the number of ticks of all life stages...”

The authors completely ignored the fact that Lake Paranoá (in Brasília) is surrounded by a highly urbanized matrix. Anyone who reads the manuscript and doesn't know Brasília might even think that the study was carried out on the edge of a peri-urban lake, which is definitely not the case. The lack of relationship between the number of capybaras and the abundance of ticks in the sampled region is most likely related to urbanization. Most likely, the number of ticks may be related more to the number of domestic animals (especially dogs and cats) than specifically to capybaras. It is true that capybaras are the primary hosts of A. dubitatum, however, the same cannot be said for A. sculptum. It has been demonstrated that A. sculptum also parasitizes dogs and cats, including ticks infected with Rickettsia (see: Mendes et al. 2019 and Campos et al. 2020). Lake Paranoá, in addition to being surrounded by luxurious mansions and farms, whose owners raise many dogs and cats, residents from other regions of Brasília take their pets to the place for leisurely walks. The presence of these domestic animals in the research region, in my opinion, should not have been ignored in the experimental design. As it is not possible to return to the data collection phase, I suggest that the authors discuss whether urbanization and the presence of dogs and cats in the sampled area would have any impact on the results achieved.

Authors: We appreciate the reviewer's comment. We have detailed the study area in lines 88-90 by clarifying that Lake Paranoá is an urban lake. However, we have not discussed the influence of domestic animals in more detail as this metric was not measured. To make the points raised by the reviewer clearer, we have modified figure 1 and added the land use around the study area, leaving no doubt about the urbanization of the region studied. However, in the discussion (lines 381-384), we emphasized that the number of A. sculptum ticks may be related to the number of domestic animals, particularly dogs and cats (Mendes et al. 2019, Campos et al. 2020), rather than specifically to capybaras due to urbanization. It is worth noting that residents of luxurious houses often take their pets for leisurely walks, which may expose them to ticks. 

It is also necessary to improve the quality of the figures. 

Authors: Done.

References:

Campos SDE, Cunha NC, Machado CSC, Nadal NV, Seabra Junior ES, Telleria EL, et al. Spotted fever group rickettsial infection in dogs and their ticks from domestic–wildlife interface areas in southeastern Brazil. Braz J Vet Parasitol 2020; 29(1): e020219. http://doi.org/10.1590/S1984-29612020012.

Mendes JCR, Kmetiuk LB, Martins CM, Canavessi AMO, Jimenez T, Pellizzaro M, Martins TF, Morikawa VM, Santos APD, Labruna MB, Biondo AW. 2019. Serosurvey of Rickettsia spp. in cats from a Brazilian spotted fever-endemic area. Rev Bras Parasitol Vet 28:713–721.

Authors: Thanks, we included these references

Reviewer #2: - IT IS A DETAILED STUDY INCLUDING INFORMATION ABOUT ABIOTIC AND BIOTIC FACTORS.

Authors: Thanks.

BUT SOME ISSUES ARE NOT CLEAR, FOR EXAMPLE, HOW MANY DAYS A MONTH WAS THE SAMPLING EFFORT? WAS IT A MONTHLY SAMPLING?

Authors: Our sampling effort was 5 - 6 days: Set/21: 5 days, Nov/21: 5 days, Feb/22: 5 days, Apr/22: 6 days, Jun/22: 5 days, Sep/22: 5 days. (Line 105)

OTHER QUESTIONS ARE MARKED DIRECTLY IN THE TEXT.

Authors: Formatting changes in the manuscript file have been highlighted in yellow

---

## [Decision Letter · Decision Letter 1]

20 Mar 2024

Effect of environmental variables on the abundance of Amblyomma ticks, potential vectors of Rickettsia parkeri in central Brazil

PONE-D-24-02067R1

Dear Dr. Gurgel-Goncalves,

We’re pleased to inform you that your manuscript has been judged scientifically suitable for publication and will be formally accepted for publication once it meets all outstanding technical requirements.

Kind regards,

Shawky M Aboelhadid, PhD

Academic Editor

PLOS ONE

Additional Editor Comments (optional):

Reviewers' comments:

Reviewer's Responses to Questions

**Comments to the Author**

1. If the authors have adequately addressed your comments raised in a previous round of review and you feel that this manuscript is now acceptable for publication, you may indicate that here to bypass the “Comments to the Author” section, enter your conflict of interest statement in the “Confidential to Editor” section, and submit your "Accept" recommendation.

Reviewer #1: All comments have been addressed

2. Is the manuscript technically sound, and do the data support the conclusions?

Reviewer #1: Yes

3. Has the statistical analysis been performed appropriately and rigorously? 

Reviewer #1: Yes

4. Have the authors made all data underlying the findings in their manuscript fully available?

Reviewer #1: Yes

5. Is the manuscript presented in an intelligible fashion and written in standard English?

Reviewer #1: Yes

6. Review Comments to the Author

Reviewer #1: I am satisfied with the responses and additions made to the text by the authors. I consider that the paper should be accepted for publication.

7. PLOS authors have the option to publish the peer review history of their article (what does this mean?). If published, this will include your full peer review and any attached files.

Reviewer #1: No

---

## [Editor Report · Acceptance letter]

3 May 2024

PONE-D-24-02067R1 

PLOS ONE

Dear Dr. Gurgel-Gonçalves, 

I'm pleased to inform you that your manuscript has been deemed suitable for publication in PLOS ONE. Congratulations! Your manuscript is now being handed over to our production team.

Kind regards, 

on behalf of

Professor Shawky M Aboelhadid 

Academic Editor

PLOS ONE